# The Effect of Built Environment on Physical Health and Mental Health of Adults: A Nationwide Cross-Sectional Study in China

**DOI:** 10.3390/ijerph19116492

**Published:** 2022-05-26

**Authors:** Jie Tang, Nanqian Chen, Hailun Liang, Xu Gao

**Affiliations:** 1School of Public Administration and Policy, Renmin University of China, Beijing 100872, China; cc@ruc.edu.cn (J.T.); nanqian_chen5097@ruc.edu.cn (N.C.); 2Department of Occupational and Environmental Health Sciences, School of Public Health, Peking University, Beijing 100871, China

**Keywords:** housing condition, neighborhood environment, social-ecologic theory, Chinese residents

## Abstract

At present, there is a lack of research examining the relationships between the built environment and health status from a social epidemiological perspective. With this in mind, the present study aimed to explore the construct validity of housing/neighborhood conditions and evaluate the associations between the built environment and self-rated health among Chinese residents. To conduct the analysis, data from 4906 participants were derived from the 2016 China Labor-force Dynamics Survey (CLDS). Specifically, we used exploratory factor analysis to identify construct of housing/neighborhood factors and performed principal component regression (PCR) to assess the relationship between the built environment and both self-rated physical health and mental health. This process identified five common factors that corresponded to the built environment, including housing affordability, housing quality, neighborhood services, neighborhood physical environment, and perceived environment. The regression results suggested that housing affordability was negatively related to health status. Meanwhile, the services, physical environment, and perceived environment of neighborhoods were related to positive health outcomes. The influence of housing on health exhibits group heterogeneity: respondents in the 41 to 65 age group were most vulnerable to poor built environments. Whilst the results indicated that housing factors and neighborhood conditions were related to health outcomes, their influence varied across different age groups. Future interventions should be intentionally designed to target housing affordability and neighborhood factors, which may include the provision of housing assistance programs and planning layouts.

## 1. Introduction

Social–ecological models indicated that in addition to individual characteristics and behaviors (e.g., genes, diet, smoking, etc.), environmental factors (physical, social, or economic) also impact personal health [1]. As such, the environment plays an important role in determining an individual’s health, especially the built environment [2,3]. The term ‘built environment’ refers to both human-made or modified objects (homes, schools, workplaces, etc.) and the physical form of a neighborhood (buildings, spaces, parks, etc.) [3]. In health-related research, housing and neighborhood conditions are regarded as key public health issues [4,5].

Research examining the links between the built environment and health is well-established in developed countries. The findings of such research suggest that the built environment influences health through different pathways [6,7]. Specifically, housing conditions such as affordability [8,9], quality [10,11,12], and security [8,13,14] have been found to impact self-rated health status [10] and increase the risk of depression [15] and other diseases [9,16]. In addition, neighborhood characteristics including physical characteristics (e.g., pollution), social–cultural factors (e.g., social cohesion), and community resource access (e.g., recreational facilities) [17,18,19,20] have each been observed to impact residents’ physical [18,21] and mental health [22].

Compared with international research, issues related to the built environment and their impact on health have received little by way of academic attention in China [23,24]. Due to the acceleration of urbanization, China is currently facing a prominent housing problem, which urgently requires resolution to minimize its possible health implications. Therefore, it is of practical and theoretical significance to study the housing and health issues in the Chinese context. Moreover, studies that include the whole set of housing/neighborhood factors are needed. Research from a macro perspective is essential to better comprehend the full picture of how housing conditions impact health. To bridge the knowledge gap, we used data from the 2016 China Labor-force Dynamics Survey (CLDS) to explore the relationship between the built environment and health in the Chinese context. The structure of this paper is as follows: the second section sets out the theoretical framework and literature review. The third section describes the variable selection, data sources, and research methods used in the present study. The fourth section presents the empirical results of the analyses, and the final section offers a conclusion to the study, policy suggestions, as well as suggestions for future studies.

## 2. Theoretical Framework and Literature Review

### 2.1. Theoretical Framework

Previous studies have investigated the mechanism of action and subsequent impact of housing conditions and neighborhood environments on health [6,7]. For present purposes, these theoretical frameworks can be used to screen variables. One framework, the Index of Housing Insults (IHI), was proposed to capture the means by which housing bundles influence health and wellbeing [6]. To be specific, the framework includes five domains of housing/neighborhood conditions: affordability, security, quality of dwelling, quality of residential areas, and access to services.

Previous studies found that the physical and perceived environments of neighborhoods also contributed to health [18,19,25,26,27]. We adapted the theoretical framework and included the following domains: housing affordability, housing security, housing quality, neighborhood services, physical environment, and perceived environment.

### 2.2. Literature Review

Evidence has been published highlighting how housing is related to self-rated physical and mental health. Biological studies identified various mechanisms of action such as C-reactive protein (CRP), which is a biomarker associated with infection and stress. In fact, housing tenure and cost burden were shown to be associated with the presence of this protein [28]. Moreover, living in poor housing conditions also increases the risk of developing a mental illness [13]. In the research on the relationship between housing and health, home and neighborhood conditions were two dimensions marked as having an impact on health.

There are several pathways that describe the effect of housing conditions on health. The first pathway was related to housing affordability. Housing is typically regarded as “affordable” when a family spends less than 30 percent of their income to rent or purchase a residence. Associated studies showed that severe housing cost burdens were the most prevalent housing-related issue for low- and moderate-income families [16]. Meanwhile, renters were more likely to exhibit depressive symptoms and poor self-rated health than homeowners [8,29]. The high-cost burden of housing limits the amount of resources renters can use to attend to their healthcare needs, which may be detrimental to their long-term health [30]. In addition, struggling to pay for a mortgage may lead to evictions and property foreclosure, which again can precipitate symptoms of depression [31].

Second, the health impacts of housing security (tenure) were another risk factor relating to health. Homeowners are more likely to report a better health status. Researchers from Korea, Britain, Japan, and New Zealand have all identified a statistically significant relationship between housing tenure and health measures. In contrast with owner-occupiers, renters (either private renters or publicly subsidized renters) were more likely to report poor self-rated health and experience symptoms of depression [8,13,14,15]. For those tenants aged between 50–85 years, lower levels of quality of life and higher levels of depression symptoms were found to manifest over time [8,13,14,15].

Third, people who live in poor-quality houses were more likely to report poor health outcomes [10]. There is a growing body of evidence indicating that housing contributes to population health. At the same time, improvements in population health may not be possible without first addressing deficiencies in the housing infrastructure [10]. Housing characteristics relating to the overall quality [6], issues related to crowding [11,12], inaccessibility [11,12], very low or very high indoor temperatures [32], indoor air quality [33], and dampness [34] may adversely impact health, particularly amongst children and the elderly [33,35].

Fourth, neighborhood characteristics have also been identified as important determinants of individual health and wellbeing [36]. Researchers pointed out that three types of neighborhood characteristics may be important mediators of residents’ health, including physical characteristics (e.g., pollution), social–cultural factors (e.g., social cohesion), and community resource access (e.g., recreational facilities) [17]. Neighborhood socioeconomic structural factors, such as socioeconomic status and ethnic diversity, were associated with depression symptoms and poor self-reported health [37]. Moreover, studies have examined the relationship between activity facilities in neighborhoods and residents’ health [20,26,38]. Regular physical activity has not only been shown to improve one’s brain health and weight management, but also significantly reduce the frequency of depressive symptoms [20]. Doyle et al. (2006) suggested that individuals would be healthier if they lived in an active community environment, which they defined as one in which they could easily participate in physical activities [36]. Notably, better-quality citizen services in neighborhoods were associated with better self-reported general health and wellbeing across all age groups [19,21].

Fifth, the physical environment has been shown to influence health outcomes. Community environmental conditions include air quality, water quality, and the physical makeup of the community [39]. For example, those individuals who live near highways, factories, bus depots, power plants, and airports suffer from poor air and water quality, and as such are more likely to experience health conditions [39]. Exposure to higher concentrations of air pollution may affect cognitive function among older adults and worsen the symptoms of asthma sufferers [40,41]. It has also been observed that air pollution tends to be higher in more deprived neighborhoods [25]. It is interesting to note that evidence has emerged relating to the potential role of environmental pollutants (lead, PCBs, air pollution) in causing attention deficit hyperactivity disorder and autism [18].

Sixth, the cultural environment has been shown to be positively related to health outcomes: residents who live in more harmonious, safe neighborhoods report better health outcomes. Golant’s (2011) [42] “residential normalcy” model argued that residential comfort experiences (i.e., pleasurable or hassle-free feelings) and mastery experiences (i.e., trusting human relationships) contribute to a positive emotion-based fit. Some studies have preliminarily confirmed the relationship between the neighborhood environment (i.e., neighborhood safety, social cohesion) and health status [27,43]. Elliott et al. (2014) identified an association between the perception of neighborhood cohesion and mental wellbeing, which was stronger in adults over the age of 65 [43]. In addition, contextual neighborhood interactions, reciprocity [27], and the perception of better-quality public safety were associated with less psychological distress [19,22,44].

Based on the evidence detailed above, we drew on Emma Baker’s model and integrated the key housing factors affecting health outcomes into our theoretical framework. Although previous studies observed relationships between housing conditions and health outcomes, most studies only tested associations between individual living factors and health. As a result, there is a need for studies including the whole set of housing/neighborhood factors. At the same time, research from a macro perspective is essential to better comprehend how housing influences health. Moreover, although previous studies tended to focus more on the elderly and children, middle-aged working people were more likely to change their living area and experience adverse health status as a result of their new housing and neighborhood conditions.

## 3. Materials and Methods

### 3.1. Data Source

The data used for the analysis in the present study were derived from the 2016 China Labor-force Dynamics Survey (CLDS). The survey has been conducted by the Social Science Research Centre at Sun Yat-Sen University since 2012 (SYSU, 2012). CLDS is a large interdisciplinary longitudinal survey that examines the current situation of and changes in the Chinese labor force. It covers three levels of data, including the personal, family, and community levels. The variables collected are related to education, work, immigration, health, social participation, and other related research topics. The CLDS 2016 covered 29 provinces and municipalities in China (excluding Hong Kong, Macao, Taiwan, Tibet, and Hainan). The survey respondents were labor forces in their families aged 15–64 (as well as individuals aged 65 and over who were still in employment). The frequency distributions for all data items were reviewed to confirm that all responses were within the expected range according to the survey documentation. The original sample of CLDS 2016 included 21,086 individuals. After excluding 16,180 invalid samples with extreme or missing values, the final number of valid samples was 4906.

### 3.2. Exposure and Outcome Measurements

In the present study, self-rated physical health and self-rated mental health were taken as the dependent variables relevant to health outcomes. Self-rated health (SRH) is a frequently used health indicator [45] that functions as a dynamic representation of overall health, including the patient’s knowledge of current and past medical problems, current frailty, and health changes over time [46]. In this article, respondents were asked “What do you think of your current state of physical health?”. This variable in this study was coded with values ranging from 1 to 5, indicating “very unhealthy”, “less healthy”, “generally”, “healthy”, and “very healthy”. They were also asked questions related to mental health, such as “How often have you felt depressed or depressed in the past four weeks?”. This variable was again coded with values from 1 to 5, indicating “always”, “often”, “sometimes”, “rarely”, and “never”. A higher score here indicates the individual is in better health.

The independent variables were related to housing conditions and neighborhood conditions. Housing/neighborhood conditions were coded as follows (see Appendix A
Table A1): The variables related to housing conditions included housing debt (coded as yes or no), housing expenditure burden (money spent on housing and renovations last year, coded as a continuous variable), housing tenure (coded as yes or no), living space (coded as a continuous variable), and running water supply (coded as yes or no). Additionally, the built environment was measured using a perceived (self-reported) measure [47].

The variables related to neighborhood conditions included air pollution (coded from 1 to 4, a higher score means the pollution was more severe), noise pollution (coded from 1 to 4, a higher score means the pollution was more severe), soil pollution (coded from 1 to 4, a higher score means the pollution was more severe), sports place (coded as yes or no), elderly activity room (coded as yes or no), community park (coded as yes or no), neighborhood familiarity (coded from 1 to 5, a higher score means the neighborhood was more familiar), neighborhood trust (coded from 1 to 5, a higher score means the neighborhood was more trustable), mutual help (coded from 1 to 5, a higher score means the neighborhood was more supportive), and neighborhood safety (coded from 1 to 4, a higher score means the neighborhood was safer).

Demographic covariates included age (coded as a continuous variable), gender (coded as male or female), income (coded as a continuous variable), marriage (coded as married or other marriage status), residence type (hukou) (non-agricultural or agricultural), region (coded as east, middle or west), and living areas (coded as urban or rural), and work status (coded as employed or unemployed). Income was calculated as the sum of the income of all family members in 2015. Note that we took the logarithm in the following analysis. Finally, education was measured as a categorical variable and assigned a value from 1 to 3 (primary school (1), junior high school (2), high school and above (3)).

### 3.3. Statistical Analysis

As noted above, the final sample size was 4906. We conducted cross-sectional statistical analyses while accounting for the complex sampling design of the CLDS survey. We first described the sociodemographic and health-related characteristics of the survey respondents before calculating the mean values or proportions for each dependent variable. Design-based F tests were used to determine whether the means or proportions for each dependent variable between Medicaid patients and uninsured patients were statistically different. Subsequently, we conducted the principal component analysis to extract several common factors relating to housing and neighborhood dimensions. In terms of robustness testing for factor analysis, we randomly divided the total sample into two groups: we conducted Exploratory Factor Analysis (EFA) on one group and Confirmatory Factor Analysis (CFA) on the other. Finally, principal component regression (PCR) analyses were conducted to examine the relationship between housing conditions and health status. Principal Component Regression (PCR) combines PCA and OLS. Based on principle component analysis (PCA), the regression analysis (PCR) adopted the principal components (PCs) as independent variables and integrated them into the models according to the theoretical model to make an appropriate estimation of the parameters [48,49]. This approach has the advantage of improving understanding of the relationship between the built environment and health outcomes [49]. We also performed subgroup analyses to assess the robustness of the findings. It should be noted that all analyses were conducted using the STATA software, version 15.0, and two-tailed *p*-values less than or equal to 0.1 were considered statistically significant.

## 4. Results

### 4.1. Descriptive Statistics

As can be seen in Table 1, the average self-rated physical health score in the sample was 3.7, indicating that the average health status amongst all respondents was between the general and healthy levels. The average mental health score was 4.3 (ranked on a scale of 1 to 5), which indicated that most respondents had a low level of mental distress. Among the 4906 respondents, 20.3% of people reported they had housing debt, and the ratio of annual average housing expenditure was 3.8%. Approximately 87.3% of respondents had full ownership of their property. Living space per capita was 28.45 square meters and most respondents had access to running water. Regarding neighborhood conditions, the respondents indicated they were generally less exposed to air, water, noise, and soil pollution. About 67.8% of the respondents’ neighborhoods had sports facilities, 58.4% had an elderly activity room, and 43.6% had a park. Moreover, the residents were inclined to report good familiarity (3.75), mutual trust (3.65), mutual help (3.34), and safety in their neighborhoods.

Table 2 shows the distribution of the physical health status (a) and mental health status (b) of the 4906 participants by each individual, housing, and neighborhood level. In terms of physical health score (see part a of Table 2), there is significant variance across most sociodemographic groups (*p* < 0.05), socioeconomic groups (*p* < 0.1), and housing conditions (*p* < 0.01). In the neighborhood level context, significant differences were identified in the measures, except those relating to the availability of sports facilities and mutual help. In terms of mental health score (see part b of Table 2), there were no significant differences in the measures of age group and hukou. In the housing level context, we only observed mental health status variance in the housing debt (*p* < 0.01) variable. In the neighborhood level context, measures of air pollution (*p* < 0.1), water pollution (*p* < 0.1), noise pollution (*p* < 0.05), soil pollution (*p* < 0.01), mutual familiarity (*p* < 0.1), mutual help (*p* < 0.01), neighborhood safety (*p* < 0.01), and park facilities (*p* < 0.01) exhibited significant variances in mental health status.

### 4.2. Factor Analysis

We initially identified 16 items related to housing factors based on the conceptual framework, which are listed in Table 1. We used principal component analysis to identify the construct of housing factors. The Kaiser-Meyer-Olkin (KMO) was calculated to be 0.79, which exceeds the recommended minimum value of 0.60 [48,49]. Bartlett’s test of sphericity was significant (χ^2^ = 16,900.37; df = 120, *p* < 0.001) [50], indicating both that there is a strong relationship amongst the variables and also that the data were suitable for principal factor analysis. Meanwhile, Cronbach’s α was 0.678, which confirmed the reliability of these items.

Principal components analysis was then carried out to determine whether the 16 items could be combined into separate components reflecting different aspects of housing/neighborhood conditions. A five-component solution was extracted using the rule that eigenvalues are greater than 1.0. Promax rotation was then performed to minimize the complexity of the loadings for each component. Table 3 showed the eigenvalues of Factor 1 to Factor 5. A five-factor model was then employed, which accounted for 59% of the total variance. Based on the CFA, the results indicated that the model was supported by the following indices: RMSEA was 0.048 (<0.05), SRMR was 0.023 (<0.05), TLI was 0.909 (>0.9), and CFI was 0.929 (>0.9). In order to validate the model, we also performed analyses by using the Fornell and Lacker criterion and heteotrait-monotrait (HTMT). The Fronell-Larcker criterion is one of the most popular techniques used to check the discriminant validity of measurement models. This criterion suggests that the square root of the average variance extracted (AVE) by a construct must be greater than the correlation between the construct and any other construct [51]. Notably, every construct in the present article satisfied this requirement.

We also used the HTMT criterion to assess discriminant validity. The heterotrait-monotrait ratio (HTMT) of the correlations is the average of the heterotrait-heteromethod correlations, relative to the average of the monotrait-heteromethod correlations [52]. In the present study, the HTMT value was below 0.85, which indicated discriminant validation between the five factors.

Component 1 represented four items pertaining to a neighborhood’s physical environment (air pollution, water pollution, noise pollution, and soil pollution). Component 2 represented four items pertaining to a neighborhood’s perceived environment (including mutual familiarity, mutual trust, mutual help, and safety). Component 3 represented three items pertaining to concerns about the services a neighborhood offers (including sports places, elderly activity rooms, and parks). Component 4 represented three items pertaining to housing quality (including housing tenure, per living space, and running water). Finally, component 5 represented two items pertaining to housing affordability (housing debt and housing expenditure burden). The coefficients for the items are detailed in Table 4.

### 4.3. Full Sample Analysis

We performed PCR analyses to identify the significance of housing factors and health. Table 5 provides the total sample analysis results. Housing affordability was negatively related to self-rated physical health (OR = 0.945, 95% CI: 0.896,0.996; *p* < 0.05) and mental health (OR = 0.925, 95% CI: 0.876,0.976; *p* < 0.01). However, we could not discern the significant impact of housing quality on health status in this model. Elsewhere, we found that respondents living in neighborhoods with health services (such as parks and exercise facilities) were more likely to report a good physical health status (OR = 1.078, 95% CI: 1.015, 1.141; *p* < 0.05) and mental health status (OR = 1.056, 95% CI: 0.992, 1.124; *p* < 0.1). Meanwhile, the physical environment was positively related to physical health (OR = 1.100, 95% CI: 1.041, 1.164; *p* < 0.01) and mental health outcomes (OR = 1.131, 95% CI: 1.068, 1.199; *p* < 0.01), whilst the perceived environment was significantly associated with residents’ physical health (OR = 1.447, 95% CI: 1.361, 1.541; *p* < 0.01) and mental health (OR = 1.226, 95% CI: 1.151, 1.306; *p* < 0.01).

In terms of control variables, respondents who were younger, male, and married were more likely to report good self-rated physical health and mental health. Additionally, a higher income level was found to be related to better health outcomes, and education was related to good physical health and significantly related to suffering from depression. Moreover, being employed was positively related to better physical health. People living in West China were more likely to report a negative mental health status. Notably, respondents who lived in the urban areas tended to report better health outcomes compared to respondents living in rural areas.

### 4.4. Sub-Group Analysis by Age

Our sensitivity analysis explored whether changes in health status under different housing/neighborhood conditions were alike across various age groups. We divided the respondents into four age groups: aged 15 to 28, aged 29 to 40, aged 41 to 65, and aged 66 and over. Figure 1 and Figure 2 illustrate the impact of housing on self-rated health and mental health among different age groups, respectively.

In terms of physical health, housing quality and perceived environment conditions were only associated with respondents’ health status in the aged 15 to 28 group (Figure 1a). For respondents aged 29 to 40 (Figure 1b), those respondents living in neighborhoods with polluted environments were more likely to report a poor health status. For respondents aged 41 to 65 (Figure 1c), neighborhood health services, physical environment, and perceived environment were found to be significantly positively related to their health. Respondents aged 66 and over (Figure 1d) were sensitive to a better-perceived environment, although a better-perceived environment contributed to elevated health statuses across all four of the age groups.

With regard to mental health, housing affordability burden was negatively associated with better mental health amongst respondents aged 29 to 40 (OR = 0.892, 95% CI: 0.803, 0.991; *p* < 0.05), whereas a better neighborhood physical environment was positively related to good mental health (OR = 1.220, 95% CI: 1.079, 1.379; *p* < 0.01) (Figure 2b). For middle-aged respondents (41–65), housing affordability was negatively related to mental health (OR = 0.895, 95% CI: 0.832, 0.962; *p* < 0.01), whilst neighborhood physical environment was positively associated with mental health (OR = 1.111, 95% CI: 1.031, 1.198; *p* < 0.01) and perceived environment was related to better mental health (OR = 1.287, 95% CI: 1.184, 1.399; *p* < 0.01) (Figure 2c).

## 5. Discussion

Despite the increasing academic interest in the built environment (housing and neighborhood conditions) as a determinant of health, very few academics have investigated this topic in China, especially by assessing nationwide representative samples. To bridge the knowledge gap, the present study explored the associations between the built environment and health outcomes amongst Chinese respondents by employing evaluation measures with strong construct validity.

Social–ecological theory posits that environmental factors are linked to personal health [1,11]. Specifically, the built environment, such as housing and neighborhood conditions, has a large impact on an individual’s health [2,3]. Based on the theory and the framework of the Index of Housing Insults, this study has produced a number of noteworthy findings. Although we could not observe a significant effect of housing quality on health, the results suggest that housing affordability is negatively related to health outcomes. In terms of the impact of housing affordability on residents’ health, previous evidence indicated that health can be detrimentally influenced if too much of a family’s budget is committed to fixed housing costs, such that insufficient resources are available to cover medical care, transportation, and recreation [7]. In addition, we found middle-aged people experienced more emotional distress and depression than other age groups. A reason for this may be that they shoulder many burdens, such as being the main breadwinner in their family, raising children, and supporting the elderly [23]. However, the price of urban housing keeping rising rapidly in recent years; worryingly, the income growth of respondents has not kept pace with soaring urban house prices or rents [53]. For middle-aged respondents living in urban areas, the location of their house was shown to be linked to better education and medical resources for their family members. Our second hypothesis posits that housing quality may impact health; however, it was not verified either in the full sample case or in any of the individual sub-groups. The main reason was that the quality of housing has dramatically improved through economic development and urbanization, and as a result, the gaps in resident quality have been narrowing. As a result, the impact of housing quality on health was minimal [54].

In terms of the impact of neighborhood conditions on health, we found strong evidence that supports the existence of an association. Specifically, the results showed that access to a neighborhood’s services, physical environment, and perceived environment were positively related to health status. Our findings in this regard were in line with those put forward in previous studies. Stevenson et al. (2009) found that the provision of community resources (such as parks and physical exercise facilities) provided residents with spaces to relax after work, which helped to relieve their stress [17]. Neighborhood sports services played an important role in preventing the development of chronic diseases amongst the elderly [55,56]. The mechanism of action here is that a good environment was conducive to promoting residents’ participation in exercise by providing access to a wider range of healthy lifestyle choices [39,55,56].

With respect to the physical environment, it may have a direct or indirect adverse impact on residents’ health. For instance, long-term exposure to pollution caused a series of irreversible adverse health changes [57,58]. In particular, air pollutants seriously harm respiratory function [59], water pollution leads to liver and stomach cancer [60], and noise pollution can predicate hearing loss, disrupted sleep, and other disturbing symptoms [61].

In terms of perceived environment, the current study found that all age groups can benefit from a harmonious and safe neighborhood, as positive perceptions, neighborhood harmony, and safety can elicit positive psychological processes and relieve physiologic stress responses [59]. When living in a harmonious and safe neighborhood, residents may feel secure and confident in their communication with others and feel that they are being treated honestly and compassionately [42].

Moreover, our findings can offer policy guidance for policymakers to take action to mitigate the adverse effect of poor housing conditions on health. Deprived and vulnerable subgroups should also be targeted with policies to tackle health inequity, such as the deprivation of necessities [62,63,64]. However, due to the significant barriers of the hukou system, the dual land system, and the imperfect public housing system [7], China is in a critical stage of re-building its affordable housing policy framework to address the housing affordability challenges it presently faces. Currently, there is a pressing need to promote low-rent and public rental housing for low-income families, and to explore tiered housing security schemes, such as capped housing prices and co-ownership homes.

In terms of neighborhood services, there are discrepancies in public service between urban and rural areas in China. Through the urbanization process, people tend to move to big cities, which is known as the “Siphon Effect”. This leads to the imbalanced allocation of public resources and services [24]. There is a need to increase public venues and fitness facilities in rural areas. Moreover, many vulnerable populations living in less-developed areas are often unable to access public facilities. Therefore, it is imperative to decentralize the population of front-line and large cities in China and bridge the gap in public services and resources between urban and rural areas and regions. In terms of physical and perceived environments’ impacts on self-rated health, interventions designed to enhance community greenness and improve the coverage of green space are indispensable.

This study has both strengths and limitations. One strength of the study was its representativeness stemming from the use of nationwide individual-level data. Our study was one of the most recent national studies to assess the relationship between the built environment and health status in China, where urbanization is proceeding at an unprecedented speed and residential conditions are undergoing dramatic changes. Second, this study included the whole set of housing/neighborhood factors from a holistic perspective, and as such, it was able to show the whole mechanism of the built environment on health at a macro level. Third, we used evaluation measures with strong construct validity to study the association between the built environment and health. Principle component analysis and principal component regression analyses were conducted to improve our understanding of the relationship between housing/neighborhood conditions and health outcomes.

Nevertheless, the present study also had several limitations. First, due to the nature of cross-sectional data, this study was not able to capture temporal changes in housing and neighborhood conditions. Future studies may use panel data to trace the impact of dynamic changes in housing on health. Second, self-rated health and mental health scores were subjective measures, which inherently run the risk of self-report bias. In response to this, future studies could include objective health indicators, such as biomarkers, anthropometry, diagnoses, etc. Third, although our analyses controlled various confounding factors, due to the limitation of using a secondary dataset, we may have overlooked other factors related to the health outcomes, such as unhealthy lifestyles and the stability of residence. Finally, neighborhood conditions may be tied to factors that may influence their detection, such as better access to medical services (mental health services in particular). Although we controlled some socioeconomic variables such as income and work status, which may be related to the accessibility to medical services, we still could not deal with this potential detection bias. Future studies should adopt mixed methods of qualitative and quantitative research to improve the robustness of the results.

## 6. Conclusions

To our best knowledge, this is one of the first interdisciplinary studies integrating urban and environmental health sciences with social epidemiology. Our study included a whole set of built environment factors from a holistic perspective and utilized principal component regression analyses to improve the understanding of the relationship between housing/neighborhood conditions and health outcomes. The findings suggest that housing affordability, neighborhood services, neighborhood physical environment, and perceived environment were significantly related to health outcomes. In addition, we found that social–economic determinants, including age, gender, and income, affected health. The influence of housing on health exhibited group heterogeneities, as respondents aged between 41 and 65 were vulnerable to poor housing conditions and also experienced poorer health status. This study confirmed the relationship between housing and health in the Chinese context at the housing neighborhood levels from an empirical perspective, which may have policy implications for future housing security reforms in China.

## Figures and Tables

**Figure 1 ijerph-19-06492-f001:**
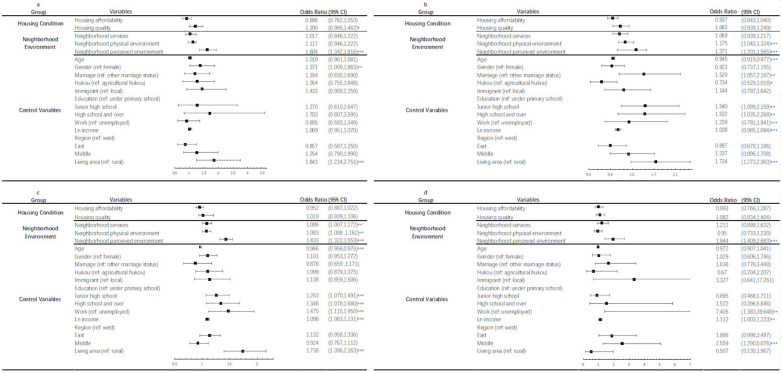
Results of the impact of built environment on physical health among different age groups: (**a**) age under 28, (**b**) age between 29 and 40, (**c**) age between 41 and 65 and (**d**) age over 66. Note: (1) 95% confidence interval in parentheses; (2) * *p* < 0.1; ** *p* < 0.05; *** *p* < 0.01.

**Figure 2 ijerph-19-06492-f002:**
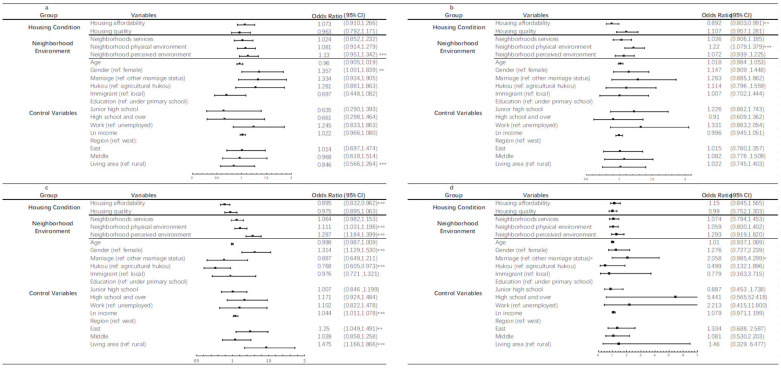
Results of the impact of built environment on mental health among different age groups: (**a**) age under 28, (**b**) age between 29 and 40, (**c**) age between 41 and 65 and (**d**) age over 66. Note: (1) 95% confidence interval in parentheses; (2) * *p* < 0.1; ** *p* < 0.05; *** *p* < 0.01.

**Table 1 ijerph-19-06492-t001:** Characteristics of 4906 participants.

Variables	Total *n* = 4906	Age Under 28 *n* = 659	Age between 29–40 *n* = 1125	Age between 41–65 *n* = 2886	Age over 66 *n* = 236	df	F	Effect Size	*p* Value
**Dependent variables**									
Physical health, mean (SE)	3.713 (0.958)	4.153 (0.747)	3.990 (0.839)	3.544 (0.979)	3.229 (0.967)	4905	138.58	0.78	0.000
Mental health, mean (SE)	4.251 (0.949)	4.223 (0.904)	4.298 (0.873)	4.236 (0.982)	4.280 (0.997)	4905	1.40	0.00	0.239
**Independent variables**									
Housing conditions									
Housing debt									
Yes	20.28	17.91	21.33	21.38	8.47	4905	8.56	0.01	0.000
No	79.72	82.09	78.67	78.62	91.53				
Housing expenditure burden, mean (SE)	0.039 (0.160)	0.037 (0.147)	0.048 (0.188)	0.036 (0.149)	0.038 (0.169)	4905	1.51	0.00	0.209
Housing tenure									
Yes	87.30	72.38	82.93	91.41	99.58	4905	79.50	0.05	0.000
No	12.70	27.62	17.07	8.59	0.42				
Living space per capita	28.450 (22.742)	25.704 (18.121)	26.116 (23.454)	29.652 (22.997)	32.538 (25.899)	4905	16.56	0.01	0.000
Running water									
Yes	83.41	90.14	88.98	81.46	61.86	4905	45.82	0.03	0.000
No	16.59	9.86	11.02	18.54	38.14				
Neighborhood conditions									
Air pollution, mean (SE)	3.124 (0.834)	3.010 (0.838)	3.019 (0.862)	3.176 (0.816)	3.301 (0.814)	4905	17.67	0.01	0.000
Water pollution, mean (SE)	3.163 (0.785)	3.073 (0.750)	3.084 (0.773)	3.208 (0.791)	3.254 (0.817)	4905	10.95	0.01	0.000
Noise pollution, mean (SE)	3.204 (0.839)	3.005 (0.883)	3.088 (0.862)	3.278 (0.813)	3.398 (0.751)	4905	32.00	0.02	0.000
Soil pollution, mean (SE)	3.368 (0.863)	3.314 (0.664)	3.311 (0.672)	3.398 (0.692)	3.415 (0.669)	4905	6.23	0.00	0.000
Sports place, %									
Yes	67.81	71.78	68.71	66.87	63.98	4905	2.64	0.00	0.048
No	32.19	28.22	31.29	33.13	36.02				
Elderly activity room, %									
Yes	58.40	65.86	65.69	55.16	42.37	4905	26.08	0.02	0.000
No	41.60	34.14	34.31	44.84	57.63				
Park, %									
Yes	43.60	52.05	45.07	41.96	33.05	4905	11.38	0.01	0.000
No	56.40	47.95	54.93	58.04	66.95				
Mutual familiarity, mean (SE)	3.748 (1.016)	3.185 (1.094)	3.535 (1.011)	3.926 (0.946)	4.161 (0.804)	4905	136.94	0.08	0.000
Mutual trust, mean (SE)	3.651 (0.849)	3.311 (0.808)	3.480 (0.816)	3.772 (0.840)	3.932 (0.807)	4905	82.48	0.05	0.000
Mutual help, mean (SE)	3.339 (1.041)	2.950 (1.044)	3.167 (1.019)	3.475 (1.016)	3.576 (1.047)	4905	63.66	0.04	0.000
Neighborhood safety, mean (SE)	3.242 (0.651)	3.107 (0.674)	2.156 (0.630)	3.301 (0.642)	3.310 (0.698)	4905	25.19	0.15	0.000

**Table 2 ijerph-19-06492-t002:** (**a**) Distribution of physical health status among 4906 participants in different sociodemographic and socioeconomic groups, housing and neighborhood conditions. (**b**) Distribution of mental health status among 4906 participants in different sociodemographic and socioeconomic groups, housing, and neighborhood conditions.

(**a**)				
**A**	**Physical Health**			
	**Mean (SE)**	**df**	**F or T ***	**Effect Size ^†^**
**Control Variables**				
Age group ***				
Age under 28	4.153 (0.747)	4905	138.58	0.08
Age between 29–40	3.99 (0.839)			
Age between 41–65	3.544 (0.979)			
Age over 66	3.229 (0.967)			
Gender **	□			
Female	3.684 (0.966)	4904	−1.99	0.06
Male	3.739 (0.949)			
Marriage ***	□			
Other marriage status	3.894 (0.904)	4904	5.41	0.15
Married	3.683 (0.963)			
Hukou ***	□			
Agricultural hukou	3.642 (0.98)	4904	−8.89	0.25
Non-agricultural hukou	3.917 (0.859)			
Immigrant ***	□			
Local people	3.69 (0.969)	4904	−4.60	0.13
Immigrant	3.884 (0.852)			
Education ***	□			
Primary school and below	3.389 (1.005)	4905	165.77	0.09
Junior high school	3.773 (0.935)			
Senior High school and above	3.977 (0.831)			
Work *	□			
Unemployed	3.641 (1.02)	4904	−1.55	0.04
Have a job now	3.719 (0.952)			
Region ***	□			
East	3.753 (0.914)	4905	5.11	0.00
Middle	3.642 (0.996)			
West	3.704 (0.99)			
Living area ***	□			
Rural	3.576 (0.99)	4904	−13.49	0.39
Urban	3.952 (0.847)			
**Housing conditions**				
Housing debt ***	□			
No	3.748 (0.946)	4904	5.00	0.14
Yes	3.578 (0.99)			
Housing tenure ***	□			
No	3.915 (0.806)	4904	5.64	0.16
Yes	3.684 (0.974)			
Running water ***	□			
No	3.446 (1.007)	4904	−8.79	0.25
Yes	3.766 (0.938)			
**Neighborhood conditions**				
Air pollution ***	□			
Very serious	3.736 (0.879)	4905	6.78	0.00
Serious	3.66 (0.95)			
Not very serious	3.783 (0.923)			
Very good	3.655 (1.002)			
Water pollution ***	□			
Very serious	3.623 (1.004)	4905	12.32	0.01
Serious	3.555 (0.959)			
Not very serious	3.792 (0.916)			
Very good	3.683 (0.994)			
Noise pollution ***	□			
Very serious	3.688 (0.904)	4905	9.52	0.01
Serious	3.777 (0.904)			
Not very serious	3.783 (0.908)			
Very good	3.632 (1.016)			
Soil pollution ***	□			
Very serious	3.812 (0.928)	4905	7.19	0.00
Serious	3.568 (1.029)			
Not very serious	3.774 (0.917)			
Very good	3.676 (0.98)			
Sports place	□			
No	3.699 (1.000)	4904	−0.74	0.02
Yes	3.72 (0.937)			
Elderly activity room ***	□			
No	3.589 (0.996)	4904	−7.72	0.22
Yes	3.802 (0.919)			
Park ***	□			
No	3.631 (0.977)	4904	−6.87	0.20
Yes	3.82 (0.921)			
Mutual familiarity **	□			
Very low	3.774 (1.017)	4905	3.19	0.00
Relatively low	3.841 (0.936)			
General	3.699 (0.911)			
Relatively high	3.677 (0.926)			
Very high	3.724 (1.044)			
Mutual help**	□			
Very low	3.553 (1.002)	4905	3.27	0.00
Relatively low	3.747 (0.915)			
General	3.679 (0.951)			
Relatively high	3.745 (0.925)			
Very high	3.729 (1.111)			
Mutual trust	□			
Very low	3.55 (0.986)	4905	1.67	0.00
Relatively low	3.698 (1.02)			
General	3.694 (0.927)			
Relatively high	3.707 (0.929)			
Very high	3.79 (1.065)			
Neighborhood safety ***	□			
Very unsafe	3.552 (0.958)	4905	8.38	0.01
Unsafe	3.7 (0.897)			
General	3.537 (0.975)			
Safe	3.782 (1.036)			
(**b**)				
**B**	**Mental Health**			
	**Mean (SE)**	**df**	**F or T ***	**Effect Size ^†^**
**Control Variables**				
Age group				
Age under 28	4.223 (0.904)	4905	1.4	0.00
Age between 29–40	4.298 (0.873)			
Age between 41–65	4.236 (0.983)			
Age over 66	4.28 (0.997)			
Gender ***	□			
Female	4.173 (0.987)	4904	−5.42	0.15
Male	4.32 (0.909)			
Marriage	□			
Other marriage status *	4.189 (0.942)	4904	−1.86	0.05
Married	4.261 (0.95)			
Hukou	□			
Agricultural hukou	4.244 (0.967)	4904	−0.85	0.02
Non-agricultural hukou	4.27 (0.897)			
Immigrant	□			
Local people ***	4.262 (0.951)	4904	2.31	0.07
Immigrant	4.166 (0.934)			
Education ***	□			
Primary school and below	4.191 (1.019)	4905	4.84	0.00
Junior high school	4.276 (0.946)			
Senior High school and above	4.284 (0.874)			
Work ***	□			
Unemployed	4.146 (1.006)	4904	−2.26	0.06
Have a job now	4.26 (0.944)			
Region **	□			
East	4.286 (0.935)	4905	3.04	0.00
Middle	4.214 (0.942)			
West	4.223 (0.974)			
Living area *	□			
Rural	4.237 (0.982)	4904	−1.37	0.04
Urban	4.275 (0.89)			
**Housing conditions**				
Housing debt ***	□			
No	4.284 (0.923)	4904	4.82	0.14
Yes	4.122 (1.035)			
Housing tenure	□			
No	4.236 (0.925)	4904	−0.41	0.01
Yes	4.253 (0.953)			
Running water	□			
No	4.22 (1.027)	4904	−1.01	0.03
Yes	4.257 (0.933)			
**Neighborhood conditions**				
Air pollution *	□			
Very serious	4.113 (1.005)	4905	2.33	0.00
Serious	4.229 (0.949)			
Not very serious	4.245 (0.929)			
Very good	4.282 (0.964)			
Water pollution *	□			
Very serious	4.192 (1.086)	4905	2.40	0.00
Serious	4.188 (0.987)			
Not very serious	4.241 (0.926)			
Very good	4.291 (0.948)			
Noise pollution **	□			
Very serious	4.103 (1.03)	4905	3.24	0.00
Serious	4.215 (0.933)			
Not very serious	4.24 (0.921)			
Very good	4.287 (0.969)			
Soil pollution ***	□			
Very serious	4.232 (1.031)	4905	4.05	0.00
Serious	4.152 (1.03)			
Not very serious	4.218 (0.925)			
Very good	4.297 (0.954)			
Sports place	□			
No	4.252 (0.94)	4904	0.07	0.00
Yes	4.25 (0.954)			
Elderly activity room	□			
No	4.222 (0.972)	4904	−1.79	0.05
Yes	4.271 (0.932)			
Park ***	□			
No	4.214 (0.976)	4904	−3.06	0.09
Yes	4.298 (0.912)			
Mutual familiarity *	□			
Very low	4.151 (0.954)	4905	2.19	0.00
Relatively low	4.188 (0.91)			
General	4.225 (0.95)			
Relatively high	4.251 (0.946)			
Very high	4.308 (0.967)			
Mutual help ***	□			
Very low	4 (1.09)	4905	8.07	0.00
Relatively low	4.284 (0.915)			
General	4.225 (0.938)			
Relatively high	4.258 (0.949)			
Very high	4.384 (0.92)			
Mutual trust ***	□			
Very low	4.05 (1.154)	4905	13.66	0.01
Relatively low	4.081 (1.002)			
General	4.171 (0.964)			
Relatively high	4.279 (0.933)			
Very high	4.435 (0.889)			
Neighborhood safety ***	□			
Very unsafe	4.034 (1.169)	4905	19.46	0.01
Unsafe	4.235 (0.925)			
General	3.973 (1.055)			
Safe	4.349 (0.936)			

Note for part a: *: Denoted by F in ANOVA, T in *t* test. ^†^: Effect size was denoted by Eta-Squared in ANOVA, Cohen’s d in T test. * *p* < 0.1; ** *p* < 0.05; *** *p* < 0.01. Note for part b: *: Denoted by F in ANOVA, T in *t* test. ^†^: Effect size was denoted by Eta-Squared in ANOVA, Cohen’s d in *t* test. * *p* < 0.1; ** *p* < 0.05; *** *p* < 0.01.

**Table 3 ijerph-19-06492-t003:** Rotated factor loadings and unique variances.

Factor	Eigenvalue	Difference	Proportion	Cumulative
Factor1	3.47577	1.42961	0.2172	0.2172
Factor2	2.04616	0.43142	0.1279	0.3451
Factor3	1.61474	0.43853	0.1009	0.4460
Factor4	1.17621	0.13552	0.0735	0.5196
Factor5	1.04070	0.13451	0.0650	0.5846

**Table 4 ijerph-19-06492-t004:** Matrix of factors.

	Factor1	Factor2	Factor3	Factor4	Factor5
Housing debt					0.7334
Housing expenditure burden					0.7852
Housing tenure				0.6111	
Per living space				0.7400	
Running water				−0.5725	
Air pollution	0.8161				
Water pollution	0.8312				
Noise pollution	0.7162				
Soil pollution	0.8367				
Sports place			0.7672		
Elderly activity room			0.7860		
Park			0.7153		
Mutual familiarity		0.8103			
Mutual trust		0.8834			
Mutual help		0.8308			
Safety		0.4310			

**Table 5 ijerph-19-06492-t005:** Ordered logistic regression results of the associations between built environment and health status.

	Model1 ^†^ Physical Health		Model2 ^‡^ Mental Health	
Variables	Odds Ratio (95% CI)	*p* ^§^	Odds Ratio (95% CI)	*p*
Housing affordability	0.945 ** (0.896, 0.996)	0.036	0.925 *** (0.876, 0.976)	0.004
Housing quality	1.050 (0.986, 1.119)	0.130	0.994 (0.930, 1.062)	0.807
Neighborhood services	1.078 ** (1.015, 1.141)	0.014	1.056 * (0.992, 1.124)	0.055
Neighborhood physical environment	1.100 *** (1.041, 1.164)	0.001	1.131 *** (1.068, 1.199)	0.000
Neighborhood perceived environment	1.447 *** (1.361, 1.541)	0.000	1.226 *** (1.151, 1.306)	0.000
Age	0.958 *** (0.953, 0.962)	0.000	0.997 (0.992, 1.002)	0.167
Gender (ref: female)	1.078 (0.977, 1.213)	0.122	1.280 *** (1.143, 1.433)	0.000
Marriage (ref: other marriage status)	1.113 (0.965, 1.326)	0.129	1.178 ** (0.999, 1.388)	0.048
Hukou (ref: agricultural hukou)	0.990 (0.830, 1.142)	0.739	0.931 (0.789, 1.100)	0.357
Immigrant (ref: local)	1.141	0.169	0.851	0.114
	(0.945, 1.384)		(0.701, 1.034)	
Education (ref: under primary school)				
Junior high school	1.324 *** (1.152, 1.521)	0.000	1.052 (0.910, 1.216)	0.494
High school and over	1.437 *** (1.211, 1.706)	0.000	1.048 (0.876, 1.253)	0.611
Work (ref: unemployed)	1.240 ** (1.018, 1.510)	0.035	1.127 (0.922, 1.378)	0.237
Ln income	1.071 *** (1.047, 1.097)	0.000	1.030 ** (1.006, 1.054)	0.012
Region (ref: west)				
East	1.052 (0.924,1.197)	0.446	1.169 ** (1.022, 1.338)	0.023
Middle	1.028 (0.888, 1.190)	0.713	1.007 (0.867, 1.170)	0.923
Living area (ref: rural)	1.712 *** (1.462, 2.006)	0.000	1.224 ** (1.038, 1.443)	0.018

Note: †: model 1 was associations between built environment and self-rated health; ‡: model2 was associations between built environment and self-rated mental health; §: * *p* < 0.1; ** *p* < 0.05; *** *p* < 0.01.

## Data Availability

Publicly available datasets were analyzed in this study. This data can be found here: http://css.sysu.edu.cn/Data (accessed on 7 September 2020).

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
