# Peer review of "The Effect of Built Environment on Physical Health and Mental Health of Adults: A Nationwide Cross-Sectional Study in China"

_ijerph, 2022, doi:10.3390/ijerph19116492_

Round 1

Reviewer 1 Report

The manuscript entitled "The Effect of Built Environment on Physical Health and Mental Health of Adults: A Nationwide Cross-Sectional Study in China" is a fascinating study on the associations between housing conditions and physical and mental health. The main strength of this study is the development of a new tool to measure housing conditions. Although the design of the study is appropriate, some suggestions may improve the study:

  1. Table 1 is informative, but I suggest adding some statistics for age group differences in the last few columns: ANOVA or Pearson's Χ2 statistic (dependent on the data type: categorical or continuous), p-value, and effect size (e.g., η2 or Cramer's V, respectively) in each row.
  2. Please add some more statistics to examine the construct validity (i.e., convergent and divergent validity) of the current factor model, such as reliability (e.g., Cronbach's α or ω), composite reliability (CR), average variance extracted (AVE), Fornell and Lacker criterion, or heteotrait-monotrait (HTMT).
  3. It is suggested to randomly divide the total sample into two groups and to conduct EFA (PCA) in one group, whereas CFA in the second group. The fit indices should be reported for the model.
  4. Figures are unclear for non-Chinese readers, so text within pictures should be translated in English.

Author Response

Thanks much for the comments and suggestions. Please see the attachment.

Reviewer 2 Report

As there were no line numbers in the manuscript, I added the comments directly into the pdf document. 

Author Response

(The authors gave the same response as above.)

Reviewer 3 Report

The paper by Jie Tang et al. entitled “The Effect of Built Environment on Physical Health and Mental Health of Adults: A Nationwide Cross-Sectional Study in China” is an interesting study on the relationship between housing/neighborhood conditions and health status from the viewpoint of epidemiology. The authors analyzed factors to identify the construct of housing/neighborhood factors. They investigated PCR to assess the relationship between housing/neighborhood conditions and health outcomes. They found five common factors. They also reported the influence of housing on health on health has group heterogeneity. They concluded that the housing factors were related to health outcomes and the influence of the built environment on health varied among different age groups. The study is well conducted. However, there are often English problems in this article, which should be proofread by a native speaker. 

  • P.1 Author profile: There are two “Correspondence:” one of which should be deleted.
  • P.1 Abstract: The number of words is 300, which should be about 200 according to the template. “(1) Background, (2) Methods, (3) Results, and (4) Conclusion” should be deleted.
  • P.2 Introduction 3rd paragraph: “health has not received” should be revised to “health have not received.”
  • P.2 2.2. 1st paragraph: “increased” should be revised to “increases.”
  • P.3 2.2. 2nd paragraph: “lead to be evicted and face” should be revised to “lead to being evicted and facing.”
  • P.3 2.2. 5th paragraph “pointed that” should be revised to “pointed out that.” “such” should be revised to “such as.” “participate” should be revised to “participate in.”
  • P.3 2.2. 7th paragraph: outcomes, residents lived” should be revised to “outcomes. Residents who lived”
  • P.4 2.2. 8th paragraph: “that including” should be revised to “that include.”
  • P.4 2.2. 9th paragraph: “the last was” should be revised to “the last is.” The contents of the article should move to "1. Introduction" with the research objectives described.
  • P.4 3.1. 1st paragraph: “was” should be revised to “has been.”
  • P.4 3.2. 1st paragraph: “reply question” should be revised to “reply to question.”
  • P.5 3.3. 1st paragraph: “3.3” should be revised to “3.3.” “Principle” should be revised to “Principal.” “It has advantages to improve” should be revised to “It has the advantages of improving.”
  • P.5 4.1. 1st paragraph: “sore” should be revised to “score.”
  • P.5 4.1. 2nd paragraph: “square meter” should be revised to “square meters.”
  • P.7-8 4.2. 1st paragraph: “in the Table 1” should be revised to “in Table 1.” “Joliffe, 2002” should be revised to “[46].” “Tran et al., 2018” should be revised to “[47].”
  • P.8 4.2. 2nd paragraph: “item” should be revised to “items.”
  • P.9 4.3. 1st paragraph: “Table 4 showed the results of from the total” should be revised to “Table 4 shows the results of the total.”
  • P.10 4.4. 1st paragraph: “was” should be revised to “were.”
  • P.11 Figure 1 and Figure 2 are too small to read.
  • P.12 5. 2nd paragraph: “Specially” should be revised to “Especially.”
  • P.13 “5. Conclusion” should be revised to “6. Conclusion.”

Author Response

(The authors gave the same response as above.)

Reviewer 4 Report

Comments to the author

Thank you very much for submitting this manuscript for review.

This manuscript, titled as “The Effect of Built Environment on Physical Health and Mental Health of Adults: A Nationwide Cross-Sectional Study in China”, was intended to explore the construct validity of the housing/neighborhood conditions, and to evaluate the associations between housing/neighborhood conditions and self-rated health among Chinese residents.

Major comments:

  1. As authors did not show the outcome distribution by each explained variables, however, it would be interested to know the baseline characteristics of population and number and rates of outcomes (physical health and mental health) by each neighborhood level context and in individual variables. We could expect that there could be a few outcomes observed in some groups of variables, which contributed to the greatest statistical significance, but shown a wide confidence intervals, this should be discussed.
  1. There is no validity data for using administrative data for the ascertainment of self-report health (physical health and mental health). The authors should describe any work/effort validating the information source. How accurate is the information in this database, how complete are the data.
  1. Interaction between neighborhood conditions and individual characteristics is need. If they were significant, I would suggest that you report them and explain in more detail why they were not meaningful. In any case, it is important to discuss interaction effects (or the absence thereof) in the discussion.
  2. One of the limitations is that the data does not include information on traditional risk factors including smoking, excessive alcohol drinking, and other comorbidities, that may be related to the risk of outcomes. This should be, however, included in the discussion as a limitation.
  3. How stable are the neighborhood conditions under consideration? Was there any information available about the amount of time participants resided at the coded addresses, or if they moved at some point across the study period?
  4. Of particular concern is that neighborhood conditions may be tied to factors that may affect detection, such as better access to medical services, and in particular mental health services. Could the authors provide some information to address concerns of potential detection bias, such as information about the distribution of medical access by levels of neighborhood economics? If not, I would suggest discussing this potential bias in the discussion and indicating how this bias may have affected results.

Minor comments:

  1. Figures were unreadable, please upload clear figures.
  2. Please have this paper edited by a proficient, native English speaker or, preferably, by one of the many science editing services now available.
  3. Please state ethical considerations.

Author Response

(The authors gave the same response as above.)

Round 2

Reviewer 1 Report

Unfortunately, authors do not revise the manuscript according to suggestions.

  1. I must repeat the comments: please add more statistics for age group differences in the last few columns. If this is impossible because of restriction of place to show table, these statistics must be included in the text: for ANOVA: F, df, p, effect size (e.g., η2). For Pearson's chi-square test: Χ2, df, p, and effect size (e.g., φ or Cramer's V, respectively, to the number of categories for comparison).
  2. Table 2 needs more statistics. If it is difficult, please divide the results into two tables (separate for physical health and mental health) or implement these statistics in the text for each comparison. If ANOVA was used, add F, df, effect size (e.g., η2), beside p-value.
  3. Cronbach's Alpha is below 0.70 which indicated poor reliability. Furthermore, reliability should be calculated for each component identified in the EFA and CFA. I still strongly recommend using composite reliability (CR), average variance extracted (AVE), Fornell and Lacker criterion, and heteotrait-monotrait (HTMT), to validate the model. The convergent and discriminant validation is necessary to explain the structure found in EFA and confirmed in CFA.
  4. RASEA = this is a mistake (replace with RMSEA).

Author Response

Thanks for your comments and suggestions. Please see the attachment.

Reviewer 3 Report

I reviewed the revised version of your paper. I confirmed the modification. Thank you. I have the following minor concerns.

1 ) In the abstract, you do not have to write four headings, such as (1) Background, (2) Methods, (3) results, and (4) Conclusions, as described in the template.

2) Figure 1 and 2 still cannot be seen. 

3) The native proofread will be necessary for the appropriate use of articles, commas, prepositions, and verbs tense in the article. 

.

Author Response

(The authors gave the same response as above.)
